neuroscience, evolution, ecology

brain, marsupials, imputations, Bayesian, comparative, phylogenetic

**Author for correspondence:**
Orlin S. Todorov
e-mail: o.s.todorov@uq.edu.au

# Testing hypotheses of marsupial brain size variation using phylogenetic multiple imputations and a Bayesian comparative framework

Orlin S. Todorov[1], Simone P. Blomberg[1], Anjali Goswami[2,3], Karen Sears[4], Patrik Drhlík[5], James Peters[6] and Vera Weisbecker[1,7]

[1]School of Biological Sciences, The University of Queensland, St Lucia, Queensland, Australia
[2]Genetics, Evolution, and Environment Department, University College London, UK
[3]Department of Life Sciences, Natural History Museum, London, UK
[4]Department of Ecology and Evolutionary Biology, College of Life Sciences, University of California Los Angeles, CA, USA
[5]Faculty of Mechatronics, Informatics and Interdisciplinary Studies, Technical University of Liberec, Czechia
[6]Department of Animal Biology, University of Illinois at Urbana Champaign, USA
[7]College of Science and Engineering, Flinders University, Australia

(iD) OST, 0000-0002-0295-7557; KS, 0000-0001-9744-9602; VW, 0000-0003-2370-4046

Considerable controversy exists about which hypotheses and variables best explain mammalian brain size variation. We use a new, high-coverage dataset of marsupial brain and body sizes, and the first phylogenetically imputed full datasets of 16 predictor variables, to model the prevalent hypotheses explaining brain size evolution using phylogenetically corrected Bayesian generalized linear mixed-effects modelling. Despite this comprehensive analysis, litter size emerges as the only significant predictor. Marsupials differ from the more frequently studied placentals in displaying a much lower diversity of reproductive traits, which are known to interact extensively with many behavioural and ecological predictors of brain size. Our results therefore suggest that studies of relative brain size evolution in placental mammals may require targeted co-analysis or adjustment of reproductive parameters like litter size, weaning age or gestation length. This supports suggestions that significant associations between behavioural or ecological variables with relative brain size may be due to a confounding influence of the extensive reproductive diversity of placental mammals.

## 1. Introduction

Brain size relative to body size is extremely variable across vertebrates [1,2], with mammals having both an extremely oversized brain for their body size and substantial variation within their clade. Evolutionary increases in relative mammalian brain sizes ('brain size' herein) are widely considered to arise from selection for larger brains [3,4], under the assumption that this confers better cognitive abilities and therefore greater fitness [5]. However, the kind of cognition targeted by selection has been a matter of extensive debate and has been researched using a large diversity of approaches. Three explanatory frameworks—social, ecological and cognitive—roughly summarize different schools of thought about brain size evolution [6–12]. The 'social brain' hypothesis suggests that an increase in social complexity (such as social or foraging group size and mating system) can select for larger brain sizes, and specifically larger neocortex size [13], because social interactions can be computationally complex. On the other hand, the 'ecological brain' hypothesis focuses on cognitive demands related to ecological factors (diet, home range, predation pressure) [7,11] because of the individual costs of dealing with these pressures. Lastly, the cognitive buffer hypothesis is a much more general

hypothesis regarding the evolution of brain variation, which does not associate relative size increase with particular behavioural parameters. Rather, it posits that larger brains generally improve fitness and survival, due to advantages related to negotiating novel or unpredictable environments and situations [6,14]. This 'buffer' function of the brain could generate positive feedback processes accelerating brain size evolution [14].

The debate about which of the three hypotheses best explains brain size evolution coincides with controversy over what specific variables select for the evolution of larger brains. This situation is exacerbated by poor data availability for many important variables, particularly behavioural data, such that only small subsets of species have a complete collection of variables and therefore confidence in the analyses is low [15–18].

In addition, it is widely recognized that relative brain size is probably antagonized by the high expense of brain growth and maintenance. Among other constraints, reproductive parameters and energetic maintenance are probably particularly important [19–25]. Because all selection-based hypotheses generally invoke traits tied to reproduction, it is difficult to dissect energetic reproductive effects from a selection [26,27] in cases where relative brain size is associated with a selection-based but reproduction-confounded variable. For example, home range and social group sizes are related to mating systems [28]; social group sizes are related to predation pressure [29], which in turn is highly correlated with reproduction and maternal investment [30]; energy availability for both maternal investment and maintenance is dependent on the ecological factor of diet [7].

The confounding of reproductive investment and selection agents on larger brains poses a particular problem for research into placental mammals, which attract most research interest because humans and other large-brained mammals belong to this clade. Placentals have highly varied life histories, with variation along the neonatal maturity spectrum (e.g. altricial versus precocial) being particularly implicated in the evolution of mammalian brain size [31]. By contrast, the sister radiation of placentals—the marsupials—does not have an altricial-precocial spectrum; neonates are all born at early developmental stages after a short gestation period (12–30 days) and the brain develops nearly entirely postnatally in all species [32]. In addition, placentals display widely differing types of placentation, gestation lengths and milk composition, which may increase the risk of confounding constraints of reproduction with selection on behavioural and ecological traits [33]. In marsupials, reproductive variation is lower: they have a three-phase lactation period that seems to be complex in its varying milk composition during lactation [34] but similar across species [35]. In addition, overall maternal investment time (pregnancy and weaning combined) is drastically more variable in placentals compared to marsupials [36]. Reproductive or developmental traits that might be associated with both socio-ecological behavioural variables *and* relative brain size are therefore minimized to lactation traits (mainly duration) and litter size [19,36,37]. Despite this, marsupials exhibit a diverse array of social and mating systems, diet types, home ranges and cognitive abilities compared to placentals [38], and are distributed in habitats with various levels of seasonality (New Guinea, Australia and the Americas). Moreover, aside from the lack of a corpus callosum, marsupial brains do not appear to differ from those of placentals in its cell-level or macromorphological organization [39,40]. This combination of relative developmental and reproductive homogeneity and ecological, behavioural and social diversity therefore makes marsupials perfectly suited for testing hypotheses about brain size evolution [19,36,37].

Previous work on marsupial brain size evolution [19], focusing on the Australasian radiation, has yielded little support for any of the main hypotheses of behavioural complexity. It instead identified reproductive constraints of litter size, which is a well-known effect across mammals and also birds [41,42]. However, this study suffered the common issue of low sample sizes for models, due to high numbers of missing values and a lack of broad phylogenetic representation—particularly with a view to American marsupials. Lastly, the study used a commonly employed statistical approach of phylogenetic generalized least squares (PGLS)—which is sensitive to topological errors in phylogeny, incompatible with a parallel analysis of multiple imputed datasets, and assumes a single mode of Brownian evolution throughout the whole tree [43].

In the current study, we expand existing marsupial brain size data by a third and use several novel analytical approaches providing the most comprehensive test of the main hypotheses of brain evolution. This involves the first use of phylogenetically informed multiple imputations (MI) through chained equations of missing data in brain size studies [18,44,45], followed by phylogenetically corrected Bayesian generalized linear mixed-effects modelling—MCMCglmm [46].

We first ask whether this more comprehensive approach improves inference for previously developed models of behavioural complexity and its relation to brain size in marsupials. We also add three additional important hypotheses of brain size evolution, namely whether play behaviour and conservation status (both cognitive buffer-related hypotheses) or hibernation (a brain maintenance-related hypothesis) are associated with brain size variation. To better understand the evolutionary patterns leading to relative brain size variation in marsupials, we compare the evolutionary models—early burst (EB), Brownian motion (BM) and Ornstein–Uhlenbeck (OU)—of relative brain size increase in the three landmasses (Australia, New Guinea and the Americas) and test whether evolutionary mode shifts had occurred as a result of invasion in a novel landmass.

## 2. Material and methods

All analyses were conducted in R [47]. The code to replicate all analyses, including all data, can be found on https://github.com/orlinst/Marsupial-brain-evo. Packages that were used for the analysis: phytools [48], caper [49], MCMCglmm [46], mulTree [50], mice [51], phylomice [52], geiger [53]. For plotting, ggplot2 [54] and hdrcde [55] were used.

### (a) Dataset

We use body mass as an estimate for body size, while endocranial volume (ECV) was used as an estimate for brain size. Data on brain volumes were obtained from measurements of ECV and were obtained from several different sources [19,39]. Most ECV volumes were obtained from Ashwell [39], which included: 472 skulls from 52 species of dasyuromorph (carnivorous/insectivorous) marsupials and the marsupial mole, 146 skulls from 14 species of Peramelemorphia (bilbies and bandicoots) and 639 skulls from 116 species of Diprotodontia (koala, wombats, gliders, possums, kangaroos and wallabies) from the collection of the Australian Museum in Sydney. Twenty-nine skulls from 16 species of Ameridelphian marsupials were from the Museums of Victoria and Queensland. We had added 62 new species of American

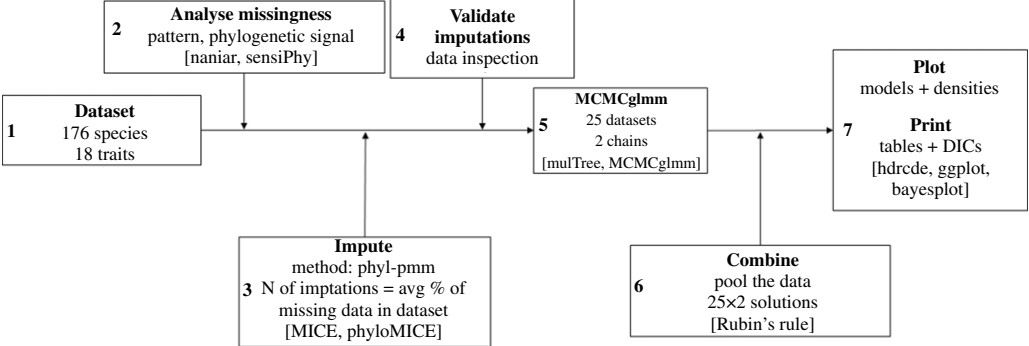

**Figure 1.** Schematic of the pipeline employed for MI and data analysis.

marsupials to the dataset, whose brain volumes were collected from museum collections using glass beads by James Peters. These data were collected in a similar way to that employed by Ashwell *et al.* [39]. Body weight data were taken from [56] but thoroughly updated using the latest data from [57]. As a result, we collated the largest and most comprehensive dataset on marsupial brain size and body weight to date comprising 176 species.

While endocranial volumes are a reliable proxy for brain size [1] they do suffer from certain drawbacks. For example, in marsupials, the koala's (*Phascolarctos cinereus*) endocranial cavity might be exceptionally large compared to the brain contained in it, comprising only around 60% of the total ECV [58]. Therefore, using ECV without correction in such species might lead to the misleading observation that they have very large brains. To our knowledge, no other species in our dataset has such a potential stark discrepancy between ECV and actual brain size.

Moreover, the dataset includes 16 traits chosen to allow for testing most of the hypotheses about brain size variation (see table with data sources in the electronic supplementary material for traits and sources). The final dataset comprises 176 species of marsupials from all three continents inhabited by the infra-class. Those comprise around 53% of all marsupial species, approximated to be around 330 in total. The full dataset used can be found in the electronic supplementary material.

Brain size, body size, origin and activity cycle had no missing values, while the other traits had around 25% missing values on average (see MI section and electronic supplementary material for the pattern of the missing data).

For a detailed description and rationale for inclusion and sources of the data, see the table with data sources.

## (b) Phylogeny

We included information on phylogenetic non-independence in all our analyses using an ultrametric phylogenetic tree of 175 extant marsupial species (with exception of the extinct Thylacine) obtained from Time Tree [59]. This was deemed appropriate because the tree provided full coverage of all species investigated, using for most taxa the recent marsupial phylogeny of Mitchell *et al.* [60].

The tree had 12 branches with the length of 0 (used as a means for resolving polytomies), which due to the requirements of some of the approaches had to be resolved. We did that by adding 0.01% of the median branch length and then ultrametricized the tree again using the extension, with the package phytools [61].

## (c) Statistical methods

The framework used is presented in the schematic view in figure 1.

## (d) Multiple imputations

For imputation of missing data, we used the R package phylomice. It is an extension for the package mice [51], which allows for MI with the addition of taking the phylogenetic non-independence of the data into account. We use the method of predictive means matching (see [62,63] for a detailed description of the non-phylogenetically corrected version of the method used), a semi-parametric stochastic regression method in which a small set of candidate values (donors) is found for each missing data point based on a Brownian motion PGLS regression model, whose predicted regression score is closest to the missing value and predictions are produced as if the species comes off the root of the tree with equal probability from five donors. Because the beta coefficient values in the regression models are chosen at random from the (approximate) joint posterior distribution, such model introduces considerable stochastic variation in the imputed data, simulated by a Markov chain Monte Carlo procedure. We have imputed 25 such datasets.

This imputation method has the advantage that missing data are imputed based on several values observed elsewhere in the set, so they are usually realistic. The pattern of missing values in our dataset is reported in the electronic supplementary material. We have variables with no missing values—brain size, body size, origin and activity period—and others with more than half of the values missing, i.e. play (68% or 120 missing) and torpor (53% or 94 missing). On average, the dataset contained 25% missing values, which we used as a reference for the number of MI (see electronic supplementary material for detailed analysis on missing data—analysis of the pattern of missingness using the package naniar [64], the phylogenetic signal in missing data using the phylo.d function in caper [49], collinearity of missingness and validation of imputed datasets). Following published recommendations from White, Royston & Wood [65], the number of datasets we imputed was equal to the percentage of missing data—25.

We ran the imputations for 500 iterations each, on natural log-transformed continuous variables, and raw values of categorical variables (see strip plot of imputations). As predictors for the imputation, only traits with less than 35% missing values were used, which rendered 13 predictors in total. The convergence of the chained equations was assessed visually on the diagnostic plots of mice, using both strip plots and density plots.

All subsequent analysis conducted on variables containing missing values were done on all 25 imputed datasets, and final results were pooled from all 25 imputations using Rubin's rule [66].

## (e) Evolutionary model variation

To assess the suggestion of Weisbecker *et al.* [19] that switches to different land masses may change patterns of marsupial brain evolution (via a change in seasonality, predation, diet), we assessed if differences in the evolutionary model on brain/body mass evolution regimes occurred in Australia, New Guinea and America. To investigate if such changes (i.e. whether Brownian motion or a specific optima-driven model best explains the tip data) and particularly, if the deepest split in the marsupial tree (Ameri- versus Australidelphia) resulted in different evolutionary patterns, we

**Table 1.** Tested models with $\beta$, standard error, posterior distribution above zero and calculated mean DIC and heritability. The values of the intercept are not included and models significantly deviating from zero are highlighted in italics.

| model | $\beta$ | s.e. | posterior distribution > 0 (%) | mean DIC/phylogenetic heritability |
|---|---|---|---|---|
| developmental | | | | −245/0.981 |
| weaning age | 0.03 | 0.03 | 77.5 | |
| litter size | −0.09 | 0.05 | *95.88* | |
| environmental | | | | −259.5/0.981 |
| diurnal | 0.03 | 0.08 | 67.4 | |
| crepuscular | −0.05 | 0.04 | 9.99 | |
| shelter safety—intermediate | 0.03 | 0.04 | 80.89 | |
| shelter safety—open | 0.05 | 0.07 | 76.06 | |
| terrestrial | −0.05 | 0.04 | 13.96 | |
| diet—2 | 0.05 | 0.06 | 79.89 | |
| diet—3 | −0.07 | 0.07 | 14.41 | |
| diet—4 | −0.03 | 0.08 | 33.62 | |
| home range | 0.01 | 0.01 | 81.21 | |
| social | | | | −270.7/0.982 |
| group living | 0 | 0.05 | 47.68 | |
| parental care | −0.02 | 0.07 | 34.07 | |
| mating system | 0.03 | 0.05 | 77.07 | |
| populations density | 0 | 0.01 | 54.85 | |
| metabolic | | | | −275.5/0.982 |
| FMR | 0.04 | 0.08 | 68.95 | |
| torpor | | | | −271.3/0.982 |
| yes | −0.13 | 0.15 | 19.22 | |
| play | | | | −248.1/0.98 |
| play—2 | 0.1 | 0.18 | 70.37 | |
| play—3 | 0.08 | 0.17 | 69.36 | |
| vulnerability | | | | −278.3/0.983 |
| status—2* | 0.02 | 0.01 | *96.94* | |
| status—3* | 0.06 | 0.06 | 84.72 | |
| origin | | | | −282/0.984 |
| origin—2 | −0.03 | 0.02 | *4.74* | |
| origin—3 | −0.05 | 0.04 | 12.15 | |

investigated which model best fitted our data—BM versus OU versus EB. Best-fitting models were assessed using the function fastBM from the geiger package. It simulates trait values given known phylogeny under the assumption of one of the evolutionary modes and then compares the simulated values to the actual ones. The fit of the models was evaluated using the Akaike information criterion (AIC). BM is a type of 'random walk' model where trait values change randomly in any direction. The EB model is a time-varying version of BM, where the Brownian rate parameter ($\sigma^2$) slows down over time (i.e. random variation decreases after an early 'burst'). OU incorporates both random variation and stabilizing selection by assuming that besides 'random walk', traits evolve towards a given optimum (adaptive evolution).

## (f) Model assessment

Due to its convenient wrapper functions, we used mulTree [50] to conduct analysis using the R package MCMCglmm [46] on each of the 25 imputed datasets. We ran the MCMC for 1 000 042 iterations, with a burn in of the first 150 000 iterations, and the sampling rate of 250. All priors were set to uniform and uninformative, which assumes that all values of the parameters are equally likely. Each model was run on two chains which produced an effective sample size of at least 3000 and all converged successfully (Gelman-Rubin criterion less than 1.1). Subsequently, the results from all 50 model runs (25 datasets on two chains) were pooled using Rubin's rules [66]. A full description of the models used can be found in the electronic supplementary material. Finally, the fit of all models to explain brain size variation was compared using phylogenetic heritabilities and the deviance information criterion (DIC). The phylogenetic heritability used in phylogenetic mixed models (PMM), measures the proportion of phenotypic variance in the sample, which is attributable to heritable factors (i.e. genes), as opposed to non-heritable factors (i.e. environmental factors or measurement error) [67]. The DIC is an estimator of prediction error like the AIC, where the estimate is based on the posterior mean. Only models with substantial posterior distribution above 0, defined as at least 95% above or below 0, were selected as being significant.

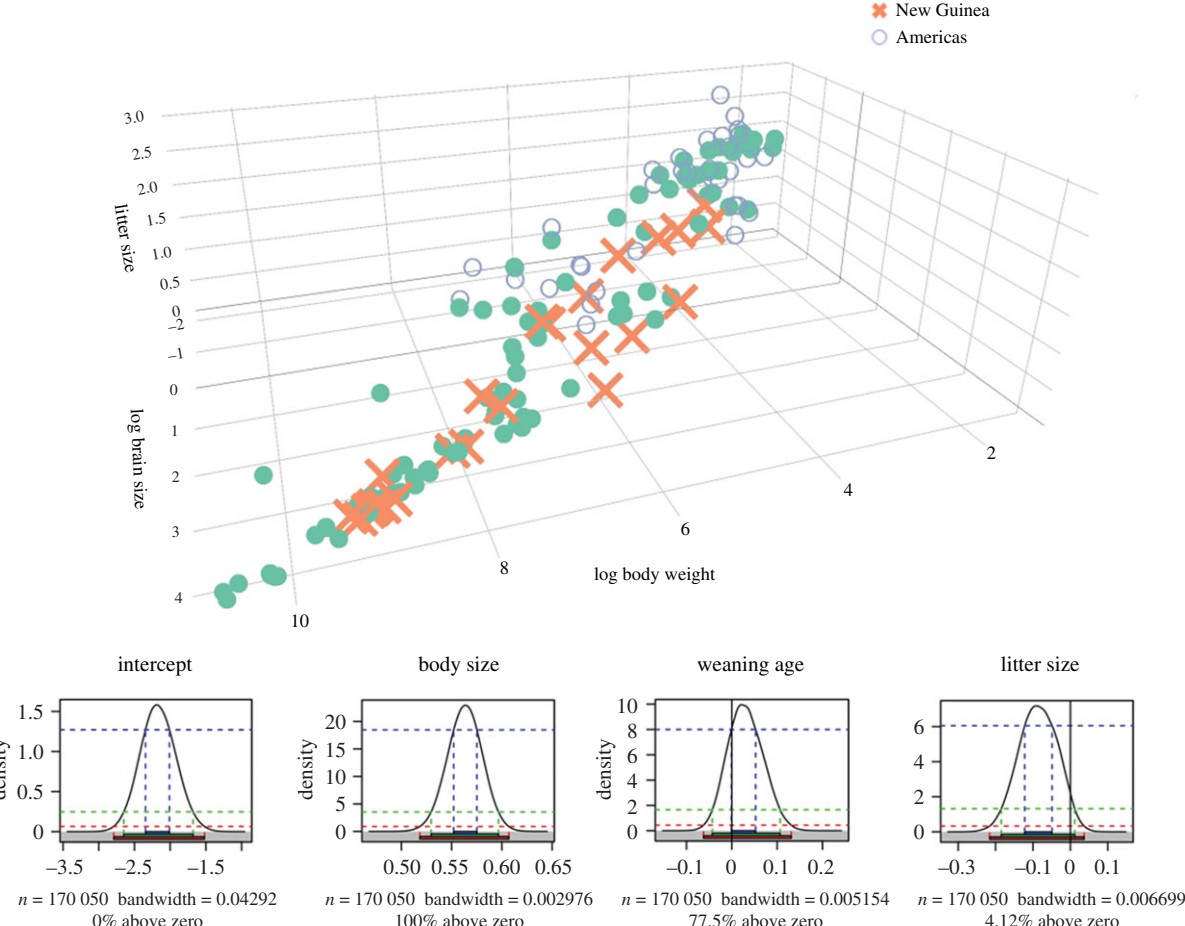

**Figure 2.** Developmental model. Three-dimensional plot of brain size, body weight and litter size including probability densities from the MCMCglmm. Note that the body weight axis is reversed. (Online version in colour.)

## 3. Results

### (a) MCMCglmm models

Eight models were investigated. In each model, body size was included as a covariate (see model description in the electronic supplementary material). *Environmental model*—Predictors: activity period, shelter safety, arboreality, diet and home range. We did not find any effect of any of the predictors on brain size. *Social model*—Predictors: group living, parental care, mating system and populations size. None of them had any effect on brain size. *Metabolic model*—The model revealed no effect of field metabolic rate on brain size, including no interaction between body size and metabolic rate. *Hibernation model*—Torpor had no effect on brain size, including no interaction between body size and torpor. *Play model*—Species with larger brain sizes did not exhibit more or more complex play behaviour compared to smaller brained species. The interaction between body size and play behaviour also did not reveal any effect of brain size. *Developmental model*—The developmental model included litter size and weaning age as predictors. Weaning age did not show a pronounced effect on brain size, but litter size had a negative effect (95.88% of the posterior distribution below zero, $\beta = -0.086$, s.e. = 0.052; figure 2). *Vulnerability model*—Vulnerable, endangered, rare, declining or species with very limited habitats were shown to have larger brains within larger bodied marsupials, but smaller brains within small bodied ones (96.94% of the posterior distribution

above zero, $\beta = 0.023$, s.e. = 0.012 for the interaction between vulnerability and body size; figure 3). *Origin model*—Species from New Guinea were shown to have larger brains within small and average body-sized marsupials, compared to Australian or American (4.74% of the posterior distribution above zero, $\beta = -0.031$, s.e. = 0.019).

We also ran a complete-case only-analysis using PGLS confirming all the results, with the exception of the developmental model, which was due to missing data included only 117 cases (see electronic supplementary material).

### (b) Evolutionary models

Our analysis of the most probable models of continuous character evolution suggested that marsupials in Australia have undergone early burst evolution of both brain and body size. By contrast, in New Guinea, we detected EB of the brain only but Brownian motion for body size evolution. In American marsupials, we determined that BM is the best fit for both brain and body size evolution. See the table of evolution models in the electronic supplementary material for more details.

A phylogenetic ANCOVA showed that a model including 'origin' as an interaction term was significantly better than a model including marsupials from all origins ($F = 5.07$, $p = 0.0072$ on 4 (full model) versus 2 (reduced model) degrees of freedom), while the variance inflation factor (VIF) was less than 2, indicating no significant multicollinearity.

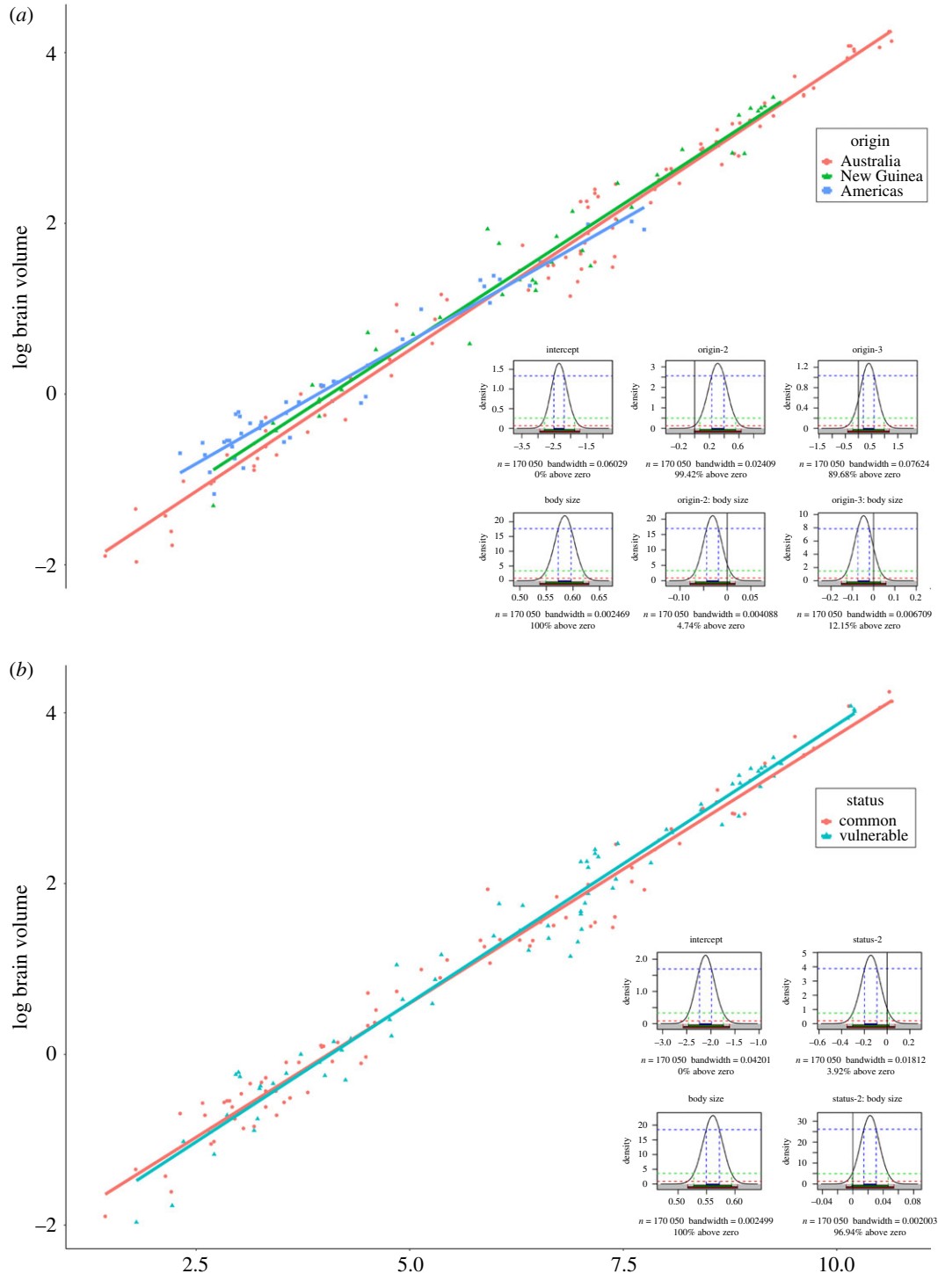

**Figure 3.** Status and origin models and probability densities from the MCMCglmm. (Online version in colour.)

## 4. Discussion

We found an intriguing lack of selection-related brain size correlates across the radiation of marsupial mammals, which we are highly confident in due to our dense phylogenetic coverage and the large datasets that our phyloMICE imputation permitted. The only unambiguous association of brain size was with litter size, which is possibly the best-known negative correlate of relative brain size across mammals [19,25,68], and beyond [20,69]. This emphasizes the high importance of reproductive investment for the evolution of relatively larger mammalian brains. Because of the conspicuous lack of association between brain size and behavioural traits that are

otherwise known to associate with brain size in placentals, our suspicion is confirmed that many of these associations may have an ultimate cause in the much more diverse range of placental developmental modes and reproductive investment [19,21,23,25,36,70–72]. At a minimum, our results demonstrate that the factors we have analysed here, and probably others, either have weak effects on relative brain size or are 'noisy' due to their high variation across radiations [19,73]. The latter conclusion is also consistent with differential models of brain and body mass evolution in marsupials of different landmasses we had detected.

Consistent with a previously established lack of association between basal metabolic rate and brain size [36], we find no

association between brain mass and field metabolic rate (a more accurate reflection of metabolic expenditure after all constituent costs are accounted for [74]). This suggests that the ongoing energetic maintenance of the brain is unlikely to be impacted by evolutionary variation in metabolic rates, at least in marsupials. It also explains why we found no relationship between hibernation/torpor and brain size, which have been hypothesized to represent times of brain starvation related to extreme temperatures [75]. However, the dataset of field metabolic rates was extensively based on phylogenetically informed estimations, rather than empirical values [76], so that a larger empirical dataset might lead to a different result.

We confirm previous reports that marsupials from New Guinea have the largest brains among marsupials, but this relationship only exists for small body masses and is mainly due to an increased intercept. This might be due to several reasons—the relationship between vulnerability and brain size (see below), or the effects of seasonality, where the more uniform, stable tropical climate in NG can facilitate the evolution of larger brains [6,77], or the effects of predation pressure in NG, both of human and non-human animals [78–80] (but see [81]).

Similar to previous studies [82,83], we did not find any clear-cut evidence that play behaviour and its complexity is related to marsupial brain size. However, our play data contained 68% imputed values, emphasizing the need for more rigorous data collection, which has been show to be related to brain size in primates [84].

We also show for the first time that larger brained marsupials are more vulnerable to extinction. This effect, again, was dependent on body size [85,86]. Vulnerable, endangered, rare, declining species or species with limited habitats had larger brains than expected, among species with larger body sizes, but smaller brains than expected among species with smaller body sizes. This is possibly because larger marsupials with larger brains tend to be more prosocial [86] and may more easily fall prey to introduced predators in areas with human activity (such as cats). On the other hand, smaller bodied marsupials with larger-than-expected brains might be more adaptable to human-modified environments due to increases in behavioural plasticity [6,14,87]. Their small sizes may facilitate the ability to avoid predation risks related to human activities and introduced predators [88]. However, the interaction between brain and body size in relation to vulnerability might again also be heavily influenced by reproductive traits. For example, preweaning predation vulnerability in placental mammmals is a major determinant of whether a species produces a few large and many small offspring within a litter, and also between a few large litters and many small ones during a reproductive season [89,90]. As such, larger-bodied marsupials, which produce smaller litters and carry their young until maturation will be at the highest risk of vulnerability.

Originally, the social brain hypothesis pointed to the relationship between cortex size and social group size in primates [13] was supported in other lineages like birds and cetaceans [9,91] and also in regard to brain substructures like the hippocampus [12]. This hypothesis had also been contested [11], and one possible reason behind the lack of relationship with social behaviour in marsupials might be the fact that in our study we used whole-brain size, and not exclusively cortex size, where the effect of reproductive constraints might be the only explanatory variable [19].

Methodologically, we were able to overcome a pervasive issue in comparative studies, namely the problem with missing data. We show that, using multiple imputation techniques and a Bayesian statistical approach, it is possible to avoid omitting whole cases due to the missingness of single datapoints. By imputing multiple datasets while conserving the mean and variance of variables with missing values and subsequently pooling the results of the statistical analysis using Rubin's rules, we were able to use the whole dataset of 176 species in all models. This is an approach that unequivocally can be useful in any comparative study, and we strongly recommend the use of the proposed pipeline and urge for further development of software tools that allow for this technique to become more widely used both with phylogenetic and non-phylogenetic data. The approach is more complicated to execute as compared to the now classical PGLS, but allows for running of stochastic models on multiple trees and datasets, and as such increases the confidence in the results.

## 5. Conclusion

Overall, our study emphasizes the possibility that many—if not most—potential explanations of relative brain size have their root in reproductive parameters, particularly those related to maternal investment. Our results also slightly favour the 'cognitive buffer hypothesis', but it is noteworthy that there is still no real clarity on what determines large brain size. There are many other, more confined and structural parameters such as (partitions, neuronal morphology and cell density) that remain unexplored and may be more important than brain size [19]; the relationship between large brains and the capacity of the skull to accommodate these is also not well-resolved and might require further study [92]. Future studies should focus on collecting more behavioural and cognitive data in the lineage in question, as this might be used not only in studies related to brain size, but also in diverse inquiries related to neuronal numbers, morphology and genetics.

Our methodological pipeline also provides a basis for an improved approach to comparative phylogenetic studies, where most tools needed for (i) phylogenetic imputations, (ii) stochastic modelling and (iii) pooling are readily available and constitute a rigorous framework for executing comparative studies.

Data accessibility. All the data and code for analysis is publicly available on the corresponding author's Github (https://github.com/orlinst/Marsupial-brain-evo), as electronic supplementary material to the paper and detailed MCMC model outputs and graphs are available from the Dryad Digital Repository: https://doi.org/10.5061/dryad.jh9w0vt9h [93].

Authors' contributions. O.S.T. conceived of the study and statistical approach, collected data, collated the dataset, analysed the data, wrote the manuscript. V.W. conceived of the study, provided data, wrote the manuscript, provided supervision to O.S.T. A.G. provided data, provided comments. K.S and J.P. provided data. S.P.B. provided comments and helped with data analysis. P.D. provided code for phylomice.

Competing interests. We declare we have no competing interests.

Funding. This study was funded by the Australian Research Council DP170103227 and FT180100634 to V.W. and a University of Queensland scholarship to O.S.T.

Acknowledgements. We thank Thomas Guillerme for help with programming. This research was carried out on the traditional lands of the Kaurna people (Flinders University), Turrbal and Jagera people (UQ). Gratitude to the Dulo and Vokil clan.

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
