## [Peer Review File · Proceedings of the Royal Society B: Biological Sciences]

Review History

RSPB-2020-2992.R0 (Original submission)

Review form: Reviewer 1

Recommendation

Major revision is needed (please make suggestions in comments)

Scientific importance: Is the manuscript an original and important contribution to its field?

Acceptable

General interest: Is the paper of sufficient general interest?

Acceptable

Quality of the paper: Is the overall quality of the paper suitable?

Poor

Is the length of the paper justified?

Yes

Should the paper be seen by a specialist statistical reviewer?

No

Do you have any concerns about statistical analyses in this paper? If so, please specify them explicitly in your report.

Yes

It is a condition of publication that authors make their supporting data, code and materials available - either as supplementary material or hosted in an external repository. Please rate, if applicable, the supporting data on the following criteria.

Is it accessible?

Yes

Is it clear?

Yes

Is it adequate?

Yes

Do you have any ethical concerns with this paper?

No

Comments to the Author

Dear authors,

I applaud your intention of showcasing the use of data imputation methods and Bayesian phylogenetic comparative analysis. I agree that these methods have the potential to improve investigations into trait evolution and that researchers should be made more aware of them and consider adding them to their tool kits. However, I think that the methodological pipeline that you propose is missing some key elements that mean that it does not currently provide the "solid basis for an improved approach to comparative phylogenetic studies" [L392] that you envisage. To inspire others to adopt these methods (and convince them of the validity of your results) you need to set an example of best practices and provide more of an orientation to the methods for people who are not familiar with them.

van Buuren (2012) provides reporting guidelines for data imputation methods. These are described in Nakagawa, S., 2015. Missing data: mechanisms, methods, and messages, in: Fox, G.A., Negrete-Yankelevich, S., Sosa, V.J. (Eds.), *Ecological Statistics: Contemporary Theory and Application*. Oxford University Press, Oxford, which is a book chapter that is well worth reading if you can access it. It provides a more extended discussion of the subject of data imputation than the Nakagawa & Freckleton (2008) paper that you cite. You have satisfied many of van Buuren's reporting guidelines, but certain key ones are missing.

Firstly, the imputation of missing data should be preceded by a detailed assessment of missingness in the data set, to understand exactly where the missing data are, whether they are clustered in any non-random way, what the mechanisms driving the missingness may be, and what biases would actually be introduced if complete case data were used. This is important so that researchers can understand exactly what effect data imputation will have on their data set, and whether it is appropriate. It may not be if, for instance, large portions of particular taxonomic groups would end up being represented only by imputed data. I do not agree with your suggestion that data imputation "is an approach that unequivocally can be useful in any comparative study" [L373] - researchers need to take steps to understand exactly what effect data imputation will have and whether it is appropriate to use on their particular data set. I would have expected to see such an assessment of missingness in your paper, along with figures showing the detailed distribution of missingness in the data set and across the phylogeny used

(to see if missingness is clustered in particular taxonomic groups), and perhaps also correlation plots showing the relationship between missingness and the variables in the data set, to see how they may be interrelated. The histogram showing overall missingness in each variable is insufficient. Perhaps the heatmap of the "Pattern" of missingness would provide more insight, but you do not provide a caption with the figure to explain what it is showing. You may find the {naniar} R package useful for conducting a detailed assessment of missingness in your data set, as well as the `miss.phylo.d` function from the {sensiPhy} package to evaluate whether missingness in the data is distributed non-randomly across the phylogeny.

Also missing was a clear description of the process and model used for the data imputation itself. The description of the process [L172-178] was opaque to me, and the model is only vaguely described when you say that you used as predictor variables the 13 variables in the data set with less than 35% missingness [L195-196]. Incidentally, by saying that you used as predictor variables the 13 variables with less than 35% missingness, does this mean that brain size was included as a predictor variable in the data imputation process? I believe that it is not good practice to use what will be the dependent variable in a subsequent analysis as part of the imputation procedure, as it could introduce circularity into analyses.

A crucial next step after imputation is to evaluate the validity of the resulting data. You say that the results are "usually realistic" [L184], but there should be a formal validation procedure. For instance, a random sample of the imputed data could be compared to qualitative descriptions of species, to check that the estimations are plausible. A sensitivity analysis should also be conducted to ensure that the imputed data were not biased by particular characteristics of the data set, phylogeny, or imputation method used. An analysis of the complete case data would also usually be presented alongside the analysis of the imputed data, so the effect of the imputed data on the statistical models and overall findings could be evaluated. Again, however, all these things are absent.

More generally, I think that you need to provide a more extended discussion of data imputation to familiarise people with the subject. In relation to this, I was surprised that you did not discuss Penone et al. (2014). Imputation of missing data in life-history trait datasets: which approach performs the best? *Methods in Ecology and Evolution* 5, 961-970, as this would seem to be important background to your argument for the adoption of data imputation methods. Such a discussion should also make people more aware of the things they should be considering when deciding whether to use data imputation in their research because, as I said above, it should not be applied unthinkingly.

As with the data imputation section, I also feel that elements are missing from your description of the MCMCglmm analyses. As with the imputation model, you do not explicitly describe the specification of the models that were analysed, and only do so informally in lines 251-264. It is not even clearly stated how brain and body size were included in the analyses, e.g., whether body size was included as a covariate or whether you used residuals from a brain-body regression. One must look in the code provided as supplementary material to find clear information about the models, but it should be front and centre in the paper.

The extra space that would be needed to add in the missing elements could be gained by removing the analysis of the mode of evolution in different radiations. Currently, this does not seem relevant to the main subject of the paper - investigating the drivers of mammalian brain evolution. It feels like it should be its own paper. If this is not appropriate, then I think it needs to be weaved together with the other analyses to a greater extent. At the moment, the only link made between the two that I can see is in lines 318-319.

My final comment is not about any missing methodological elements, but about some confusion I have about the implicit hypothesis of your study and your conclusions.

As I understand it, you say that the principal benefit of studying the causes of brain evolution in marsupials is that marsupials exhibit little variation in reproductive traits, meaning that they provide a sort of natural control that can help determine whether the correlations between socioecological variables and brain size found in placental mammals are simply due to these variables being correlated with reproductive variables, which are then in turn correlated with brain size. The hypothesis is that, if this is indeed the case, then in marsupials, which exhibit little variation in reproductive traits, there will be no correlation between reproductive traits and brain size, and this will "break" the correlation between socioecological variables and brain size. However, you find a correlation between a reproductive trait and brain size. This suggests to me that there actually is significant variation in some reproductive traits across marsupials, enough to vary strongly with brain size. The fact that you do not find correlations between socioecological variables and brain size cannot, therefore, be because the connection via reproductive traits has been "broken", because it has not. It must be for some other reason. Yet you conclude that your results show the importance of reproductive traits in mediating the effects of other variables on brain size. I do not see how this conclusion follows from your results, since your original hypothesis was not supported. Perhaps I have simply misunderstood something somewhere, but I think this apparent dissonance between your initial hypothesis and your results and conclusion needs to be clarified somehow.

In conclusion, I think that your paper has great potential to be a showcase for the use of data imputation methods and Bayesian phylogenetic comparative analysis, but I think it needs to be further fleshed out before it can become that.

More ancillary comments are listed below:

- In line 119 you say that "Data on brain volumes were derived from measurements of endocranial volumes (ECV)", and in lines 135-139 you talk about how ECV data might need correction. This suggests that you may have adjusted the ECV data in some way, but you do not explain how.
- Acronyms (e.g., BM, OU, EB) need to be introduced with the first usage of the full term.
- Variable names need to be used consistently to avoid confusion. Sometimes you use "Status" and sometimes "Vulnerability"; sometimes you use "Hibernation" and sometimes "torpor"; sometimes you use "activity period" and sometimes "diurnality", etc.
- In line 339 you say that your play behaviour data contain more than 80% imputed values, but elsewhere you say that that variable only has 68% missingness.

Review form: Reviewer 2

Recommendation

Accept with minor revision (please list in comments)

Scientific importance: Is the manuscript an original and important contribution to its field?

Excellent

General interest: Is the paper of sufficient general interest?

Good

Quality of the paper: Is the overall quality of the paper suitable?

Excellent

Is the length of the paper justified?

Yes

Should the paper be seen by a specialist statistical reviewer?

Yes

Do you have any concerns about statistical analyses in this paper? If so, please specify them explicitly in your report.

No

It is a condition of publication that authors make their supporting data, code and materials available - either as supplementary material or hosted in an external repository. Please rate, if applicable, the supporting data on the following criteria.

Is it accessible?

Yes

Is it clear?

Yes

Is it adequate?

Yes

Do you have any ethical concerns with this paper?

No

Comments to the Author

This paper uses a novel approach to find correlates of brain size evolution in marsupial mammals, by imputing missing data of predictor variables in a phylogenetic model. I cannot judge whether this method is justified or whether the imputation may lead to either exaggerating or “blurring” the patterns in available data. Perhaps an expert in statistics may help with this point. However, the materials and methods are very well described and thus the study is certainly valid. To me, it seems that there is no information added from imputation, and thus we cannot expect additional insight. But imputation may help to combine a larger number of variables in a single model. The results of such an approach should nevertheless be regarded as less reliable than those from original data.

In the current paper, I would therefore like to see the also the results of each original predictor variable (non-imputed data) in the models brain size ~ predictor * body mass, in an appendix.

Using a bayesian approach (MCMCglmm) for phylogenetic analyses is not less prone to the problem of robusticity than classic PGLS. In both methodologies, large contrasts in a variable between closely related species may have a disproportionate influence on the results, although this is less visible the more complex the analyses are designed. Thus, data quality remains of utmost importance, and the authors did a good job to compile a large sample of high quality. They even address the issue of a potential discrepancy between ECV and brain mass, which has been found in koalas, but not in any other species so far. It would be certainly interesting to study this in more species, as seasonal variation in brain size has been found e.g. in some small placental mammals.

Drawing conclusions from the analyses is a bit tricky. Actually, the negative results for any of the different realm models do not tell us much. They may stem from low power (although I don't know how to assess power in such a complicated statistical approach), or from not including covariates that are likely to be correlated with brain size, even if their effect is not reaching a significant level. But these points are mentioned in the discussion, which is well written and considers all relevant literature.

Overall, in my view this is a careful, well written study which certainly merits publication and will be of broad interest, even though it does not proclaim any flashy new findings. Its merit is the thoughtful, new methodological approach on a newly compiled comprehensive dataset. It convincingly shows that there are many unsolved questions about brain size evolution, for which insights from marsupial mammals must not be neglected.

Decision letter (RSPB-2020-2992.R0)

09-Jan-2021

Dear Mr Todorov:

I am writing to inform you that your manuscript RSPB-2020-2992 entitled "Testing hypotheses of marsupial brain size variation using phylogenetic multiple imputations and a Bayesian comparative framework" has, in its current form, been rejected for publication in Proceedings B.

This action has been taken on the advice of referees, who have recommended that substantial revisions are necessary. With this in mind we would be happy to consider a resubmission, provided the comments of the referees are fully addressed. However please note that this is not a provisional acceptance.

Sincerely,
Dr Sasha Dall
<mailto:proceedingsb@royalsociety.org>

Associate Editor
Board Member: 1
Comments to Author:

Although both reviewers agree that your manuscript has the potential to be a contribution of high impact, some issues were raised (particularly by reviewer 1) that will need to be resolved/clarified. Reviewer 1 highlights that there are key elements missing in the reporting guidelines of data imputation. He/she has kindly provided a detailed account of the issues that need to be resolved and how to resolve them. Reviewer 1 further identifies a dissonance between the hypotheses, the results, and the conclusions which may undermine the key finding of the paper. I agree with reviewer 1 that such dissonance may significantly affect the resonance of the paper and so I expect this to be fully resolved if the authors choose to resubmit.

Reviewer(s)' Comments to Author:

Referee: 1

Comments to the Author(s)

Dear authors,

I applaud your intention of showcasing the use of data imputation methods and Bayesian phylogenetic comparative analysis. I agree that these methods have the potential to improve investigations into trait evolution and that researchers should be made more aware of them and consider adding them to their tool kits. However, I think that the methodological pipeline that you propose is missing some key elements that mean that it does not currently provide the "solid basis for an improved approach to comparative phylogenetic studies" [L392] that you envisage. To inspire others to adopt these methods (and convince them of the validity of your results) you need to set an example of best practices and provide more of an orientation to the methods for people who are not familiar with them.

van Buuren (2012) provides reporting guidelines for data imputation methods. These are described in Nakagawa, S., 2015. Missing data: mechanisms, methods, and messages, in: Fox, G.A., Negrete-Yankelevich, S., Sosa, V.J. (Eds.), *Ecological Statistics: Contemporary Theory and Application*. Oxford University Press, Oxford, which is a book chapter that is well worth reading if you can access it. It provides a more extended discussion of the subject of data imputation than the Nakagawa & Freckleton (2008) paper that you cite. You have satisfied many of van Buuren's reporting guidelines, but certain key ones are missing.

Firstly, the imputation of missing data should be preceded by a detailed assessment of missingness in the data set, to understand exactly where the missing data are, whether they are clustered in any non-random way, what the mechanisms driving the missingness may be, and what biases would actually be introduced if complete case data were used. This is important so that researchers can understand exactly what effect data imputation will have on their data set, and whether it is appropriate. It may not be if, for instance, large portions of particular taxonomic groups would end up being represented only by imputed data. I do not agree with your suggestion that data imputation "is an approach that unequivocally can be useful in any comparative study" [L373] - researchers need to take steps to understand exactly what effect data imputation will have and whether it is appropriate to use on their particular data set. I would have expected to see such an assessment of missingness in your paper, along with figures showing the detailed distribution of missingness in the data set and across the phylogeny used (to see if missingness is clustered in particular taxonomic groups), and perhaps also correlation plots showing the relationship between missingness and the variables in the data set, to see how they may be interrelated. The histogram showing overall missingness in each variable is insufficient. Perhaps the heatmap of the "Pattern" of missingness would provide more insight, but you do not provide a caption with the figure to explain what it is showing. You may find the {naniar} R package useful for conducting a detailed assessment of missingness in your data set, as well as the miss.phylo.d function from the {sensiPhy} package to evaluate whether missingness in the data is distributed non-randomly across the phylogeny.

Also missing was a clear description of the process and model used for the data imputation itself. The description of the process [L172-178] was opaque to me, and the model is only vaguely

described when you say that you used as predictor variables the 13 variables in the data set with less than 35% missingness [L195-196]. Incidentally, by saying that you used as predictor variables the 13 variables with less than 35% missingness, does this mean that brain size was included as a predictor variable in the data imputation process? I believe that it is not good practice to use what will be the dependent variable in a subsequent analysis as part of the imputation procedure, as it could introduce circularity into analyses.

A crucial next step after imputation is to evaluate the validity of the resulting data. You say that the results are "usually realistic" [L184], but there should be a formal validation procedure. For instance, a random sample of the imputed data could be compared to qualitative descriptions of species, to check that the estimations are plausible. A sensitivity analysis should also be conducted to ensure that the imputed data were not biased by particular characteristics of the data set, phylogeny, or imputation method used. An analysis of the complete case data would also usually be presented alongside the analysis of the imputed data, so the effect of the imputed data on the statistical models and overall findings could be evaluated. Again, however, all these things are absent.

More generally, I think that you need to provide a more extended discussion of data imputation to familiarise people with the subject. In relation to this, I was surprised that you did not discuss Penone et al. (2014). *Imputation of missing data in life-history trait datasets: which approach performs the best?* *Methods in Ecology and Evolution* 5, 961-970, as this would seem to be important background to your argument for the adoption of data imputation methods. Such a discussion should also make people more aware of the things they should be considering when deciding whether to use data imputation in their research because, as I said above, it should not be applied unthinkingly.

As with the data imputation section, I also feel that elements are missing from your description of the MCMCglmm analyses. As with the imputation model, you do not explicitly describe the specification of the models that were analysed, and only do so informally in lines 251-264. It is not even clearly stated how brain and body size were included in the analyses, e.g., whether body size was included as a covariate or whether you used residuals from a brain-body regression. One must look in the code provided as supplementary material to find clear information about the models, but it should be front and centre in the paper.

The extra space that would be needed to add in the missing elements could be gained by removing the analysis of the mode of evolution in different radiations. Currently, this does not seem relevant to the main subject of the paper - investigating the drivers of mammalian brain evolution. It feels like it should be its own paper. If this is not appropriate, then I think it needs to be weaved together with the other analyses to a greater extent. At the moment, the only link made between the two that I can see is in lines 318-319.

My final comment is not about any missing methodological elements, but about some confusion I have about the implicit hypothesis of your study and your conclusions.

As I understand it, you say that the principal benefit of studying the causes of brain evolution in marsupials is that marsupials exhibit little variation in reproductive traits, meaning that they provide a sort of natural control that can help determine whether the correlations between socioecological variables and brain size found in placental mammals are simply due to these variables being correlated with reproductive variables, which are then in turn correlated with brain size. The hypothesis is that, if this is indeed the case, then in marsupials, which exhibit little variation in reproductive traits, there will be no correlation between reproductive traits and brain size, and this will "break" the correlation between socioecological variables and brain size. However, you find a correlation between a reproductive trait and brain size. This suggests to me that there actually is significant variation in some reproductive traits across marsupials, enough to vary strongly with brain size. The fact that you do not find correlations between socioecological variables and brain size cannot, therefore, be because the connection via

reproductive traits has been "broken", because it has not. It must be for some other reason. Yet you conclude that your results show the importance of reproductive traits in mediating the effects of other variables on brain size. I do not see how this conclusion follows from your results, since your original hypothesis was not supported. Perhaps I have simply misunderstood something somewhere, but I think this apparent dissonance between your initial hypothesis and your results and conclusion needs to be clarified somehow.

In conclusion, I think that your paper has great potential to be a showcase for the use of data imputation methods and Bayesian phylogenetic comparative analysis, but I think it needs to be further fleshed out before it can become that.

More ancillary comments are listed below:

- In line 119 you say that "Data on brain volumes were derived from measurements of endocranial volumes (ECV)", and in lines 135-139 you talk about how ECV data might need correction. This suggests that you may have adjusted the ECV data in some way, but you do not explain how.
- Acronyms (e.g., BM, OU, EB) need to be introduced with the first usage of the full term.
- Variable names need to be used consistently to avoid confusion. Sometimes you use "Status" and sometimes "Vulnerability"; sometimes you use "Hibernation" and sometimes "torpor"; sometimes you use "activity period" and sometimes "diurnality", etc.
- In line 339 you say that your play behaviour data contain more than 80% imputed values, but elsewhere you say that that variable only has 68% missingness.

Referee: 2

Comments to the Author(s)

This paper uses a novel approach to find correlates of brain size evolution in marsupial mammals, by imputing missing data of predictor variables in a phylogenetic model. I cannot judge whether this method is justified or whether the imputation may lead to either exaggerating or "blurring" the patterns in available data. Perhaps an expert in statistics may help with this point. However, the materials and methods are very well described and thus the study is certainly valid. To me, it seems that there is no information added from imputation, and thus we cannot expect additional insight. But imputation may help to combine a larger number of variables in a single model. The results of such an approach should nevertheless be regarded as less reliable than those from original data.

In the current paper, I would therefore like to see the also the results of each original predictor variable (non-imputed data) in the models $\text{brain size} \sim \text{predictor} * \text{body mass}$, in an appendix.

Using a bayesian approach (MCMCglmm) for phylogenetic analyses is not less prone to the problem of robusticity than classic PGLS. In both methodologies, large contrasts in a variable between closely related species may have a disproportionate influence on the results, although this is less visible the more complex the analyses are designed. Thus, data quality remains of utmost importance, and the authors did a good job to compile a large sample of high quality. They even address the issue of a potential discrepancy between ECV and brain mass, which has been found in koalas, but not in any other species so far. It would be certainly interesting to study this in more species, as seasonal variation in brain size has been found e.g. in some small placental mammals.

Drawing conclusions from the analyses is a bit tricky. Actually, the negative results for any of the different realm models do not tell us much. They may stem from low power (although I don't know how to assess power in such a complicated statistical approach), or from not including covariates that are likely to be correlated with brain size, even if their effect is not reaching a

significant level. But these points are mentioned in the discussion, which is well written and considers all relevant literature.

Overall, in my view this is a careful, well written study which certainly merits publication and will be of broad interest, even though it does not proclaim any flashy new findings. Its merit is the thoughtful, new methodological approach on a newly compiled comprehensive dataset. It convincingly shows that there are many unsolved questions about brain size evolution, for which insights from marsupial mammals must not be neglected.

RSPB-2021-0394.R0

Review form: Reviewer 1

Recommendation

Accept with minor revision (please list in comments)

Scientific importance: Is the manuscript an original and important contribution to its field?

Good

General interest: Is the paper of sufficient general interest?

Good

Quality of the paper: Is the overall quality of the paper suitable?

Good

Is the length of the paper justified?

Yes

Should the paper be seen by a specialist statistical reviewer?

No

Do you have any concerns about statistical analyses in this paper? If so, please specify them explicitly in your report.

Yes

It is a condition of publication that authors make their supporting data, code and materials available - either as supplementary material or hosted in an external repository. Please rate, if applicable, the supporting data on the following criteria.

Is it accessible?

Yes

Is it clear?

Yes

Is it adequate?

Yes

Do you have any ethical concerns with this paper?

No

Comments to the Author

Dear authors,

I am pleased to see that you have addressed all my previous concerns. Your overall message is now much clearer, and I appreciate how you even went beyond my suggestions when you produced the detailed profile of missingness for your data.

There are just a few minor issues remaining for me.

Firstly, I think you have misinterpreted the results of the tests for phylogenetic signal in missingness. This is completely understandable as I know that the documentation for the test function does not make it at all easy to understand how the output should be interpreted! Essentially, the most important thing to focus on is the D value: values closer to 0 indicate increasingly higher levels of phylogenetic signal, up to a perfect Brownian motion model at 0; values closer to 1 indicate an increasingly more random distribution, up to a perfectly random distribution at 1. The probability values simply indicate whether the D value is significantly different from 0 or 1. E.g., FMR has a D value of 0.09, meaning that the distribution of missingness is not random (0.09 is significantly different from 1 (random), with $p < 0.001$). FMR instead has a high phylogenetic signal, which is so high (close to 0) that it almost perfectly follows a Brownian motion distribution (0.09 is not significantly different from 0 (a Brownian motion distribution), with $p > 0.05$). Most of your variables appear to exhibit a moderate phylogenetic signal in their missingness: their D values are mid-way between 0 and 1, being both significantly different from 0 (a perfect Brownian motion distribution) and 1 (a perfectly random distribution).

Secondly, it would be better if you could format the detailed output of your PGLS models in some way - perhaps with a table for each output. It is difficult to parse the output when it is just copied-and-pasted from R.

Finally, there are some small presentation issues regarding references (L66, L77) and sentences (L76 - "variable in placentals"; L79 - "litter size in marsupials"; L138 - "collected in a similar way"; L242 - "using Rubin's rules"; L260 - "body size"). In general, I would recommend another proof-reading pass. I would also consider rewording the sentence starting on L127 - "derived" makes it sound as if you did something to the data, whereas your brain data *are* ECV.

I congratulate you on what is now a much stronger paper that should be well-received by readers!

Decision letter (RSPB-2021-0394.R0)

26-Feb-2021

Dear Mr Todorov

I am pleased to inform you that your manuscript RSPB-2021-0394 entitled "Testing hypotheses of marsupial brain size variation using phylogenetic multiple imputations and a Bayesian comparative framework" has been accepted for publication in Proceedings B.

The referee(s) have recommended publication, but also suggest some minor revisions to your manuscript. Therefore, I invite you to respond to the referee(s)' comments and revise your manuscript. Because the schedule for publication is very tight, it is a condition of publication that you submit the revised version of your manuscript within 7 days. If you do not think you will be able to meet this date please let us know.

NB. From April 1 2013, peer reviewed articles based on research funded wholly or partly by RCUK must include, if applicable, a statement on how the underlying research materials – such

as data, samples or models – can be accessed. This statement should be included in the data accessibility section.

[http://datadryad.org/submit?journalID=RSPB&manu=\(Document not available\)](http://datadryad.org/submit?journalID=RSPB&manu=(Document%20not%20available)) which will take you to your unique entry in the Dryad repository. If you have already submitted your data to dryad you can make any necessary revisions to your dataset by following the above link. Please see <https://royalsocietypublishing.org/journals/ethics-policies/data-sharing-mining/> for more details.

6) For more information on our Licence to Publish, Open Access, Cover images and Media summaries, please visit <https://royalsocietypublishing.org/journals/authors/author-guidelines/>.

Sincerely,

Dr Sasha Dall

mailto:proceedingsb@royalsocietypublishing.org

Associate Editor

Comments to Author:

We thank you for addressing all reviewer comments. In addition to the novel results, I agree that the manuscript sets an improved standard for phylogenetic comparative studies in this field. A few minor issues remain, however, that we would like you to address. Reviewer 1 points towards a possibly confusion with regards to the interpretation of the tests for phylogenetic signal in missingness, as well as some minor formatting issues.

Reviewer(s)' Comments to Author:

Referee: 1

Comments to the Author(s).

Dear authors,

I am pleased to see that you have addressed all my previous concerns. Your overall message is now much clearer, and I appreciate how you even went beyond my suggestions when you produced the detailed profile of missingness for your data.

There are just a few minor issues remaining for me.

Firstly, I think you have misinterpreted the results of the tests for phylogenetic signal in missingness. This is completely understandable as I know that the documentation for the test function does not make it at all easy to understand how the output should be interpreted! Essentially, the most important thing to focus on is the D value: values closer to 0 indicate increasingly higher levels of phylogenetic signal, up to a perfect Brownian motion model at 0; values closer to 1 indicate an increasingly more random distribution, up to a perfectly random distribution at 1. The probability values simply indicate whether the D value is significantly different from 0 or 1. E.g., FMR has a D value of 0.09, meaning that the distribution of missingness is not random (0.09 is significantly different from 1 (random), with $p < 0.001$). FMR instead has a high phylogenetic signal, which is so high (close to 0) that it almost perfectly follows a Brownian motion distribution (0.09 is not significantly different from 0 (a Brownian motion distribution), with $p > 0.05$). Most of your variables appear to exhibit a moderate phylogenetic signal in their missingness: their D values are mid-way between 0 and 1, being both significantly different from 0 (a perfect Brownian motion distribution) and 1 (a perfectly random distribution).

Secondly, it would be better if you could format the detailed output of your PGLS models in some way - perhaps with a table for each output. It is difficult to parse the output when it is just copied-and-pasted from R.

Finally, there are some small presentation issues regarding references (L66, L77) and sentences (L76 - "variable in placentals"; L79 - "litter size in marsupials"; L138 - "collected in a similar way"; L242 - "using Rubin's rules"; L260 - "body size"). In general, I would recommend another proof-reading pass. I would also consider rewording the sentence starting on L127 - "derived" makes it sound as if you did something to the data, whereas your brain data *are* ECV.

I congratulate you on what is now a much stronger paper that should be well-received by readers!

Author's Response to Decision Letter for (RSPB-2021-0394.R0)

See Appendix A.

Decision letter (RSPB-2021-0394.R1)

01-Mar-2021

Dear Mr Todorov

I am pleased to inform you that your manuscript entitled "Testing hypotheses of marsupial brain size variation using phylogenetic multiple imputations and a Bayesian comparative framework" has been accepted for publication in Proceedings B.

Open Access

Paper charges

Sincerely,
Proceedings B
mailto: proceedingsb@royalsociety.org

Appendix A

We appreciate all the kind words from the referee and are very thankful for the review. We are convinced that these recommendations significantly improved the quality of the manuscript. As advised by Reviewer 1, we have now fixed the two citations and the noticed misspelled/omitted words. We have updated one of the supplements with the suggested improvements – we have changed the interpretation of the D values and reformatted the PGLS output in tables.

Again, thanks for the scrutiny and all the important suggestions.

Orlin S. Todorov